# Anomaly Detection Using Autoencoder Reconstruction upon Industrial Motors

**DOI:** 10.3390/s22093166

**Published:** 2022-04-20

**Authors:** Sean Givnan, Carl Chalmers, Paul Fergus, Sandra Ortega-Martorell, Tom Whalley

**Affiliations:** 1School of Computing and Mathematical Sciences, Liverpool John Moores University, Byrom Street, Liverpool L3 3AF, UK; c.chalmers@ljmu.ac.uk (C.C.); p.fergus@ljmu.ac.uk (P.F.); s.ortegamartorell@ljmu.ac.uk (S.O.-M.); 2Central Group, Kitling Road, Knowsley Business Park, Liverpool L34 9JA, UK; tom.whalley@gocentral.co.uk

**Keywords:** anomaly detection, edge computing, autoencoder, predictive maintenance, condition monitoring, rotary machine, real-time monitoring, machine learning, data filtering, windowed data

## Abstract

Rotary machine breakdown detection systems are outdated and dependent upon routine testing to discover faults. This is costly and often reactive in nature. Real-time monitoring offers a solution for detecting faults without the need for manual observation. However, manual interpretation for threshold anomaly detection is often subjective and varies between industrial experts. This approach is ridged and prone to a large number of false positives. To address this issue, we propose a machine learning (ML) approach to model normal working operations and detect anomalies. The approach extracts key features from signals representing a known normal operation to model machine behaviour and automatically identify anomalies. The ML learns generalisations and generates thresholds based on fault severity. This provides engineers with a traffic light system where green is normal behaviour, amber is worrying and red signifies a machine fault. This scale allows engineers to undertake early intervention measures at the appropriate time. The approach is evaluated on windowed real machine sensor data to observe normal and abnormal behaviour. The results demonstrate that it is possible to detect anomalies within the amber range and raise alarms before machine failure.

## 1. Introduction

Rotary machine breakdowns are a significant problem in industry [1]. Breakdowns increase costs to the organisation in terms of compensation, reputation and fines incurred from missing production targets [2,3]. Electric rotary machines account for over 40% of all global electricity consumption [4], which demonstrates their high dependence worldwide. When rotary machines fault and cause prolonged unplanned shutdowns, this often has a detrimental effect on business continuity. The complete supply chain is affected, and the source organisation can often incur significant penalties. The use of machine maintenance can prevent this from happening. Thus, the rotary machines are routinely monitored, analysed, and maintenance is proactively performed.

There are three primary types of machine maintenance, reactive, preventative, and predictive. Each strategy has different benefits, costs and limitations. An increasing number of organisations are moving towards predictive maintenance in order to reduce costs and maintain efficiency. This approach is underpinned by a variety of different technologies, which include Internet of Things (IoT), Deep Learning (DL) and communication technology. The combination of IoT for remote data sensing and the advancements in sensor analytics are core components of the Industry 4.0 revolution. By utilising the self-monitoring aspects of Industry 4.0, machinery can be monitored in real-time for the assessment of ongoing health and the detection of imminent faults [5]. These strategies are important as mechanical breakdowns are inevitable across all industries. The aim is to mitigate the risk of avoidable catastrophic breakdowns to production and manufacturing facilities [3].

With a reactive maintenance strategy, machinery failure causes a reactive response to resolve the fault. This method is not suitable for many industrial practices as unexpected downtimes can be costly [6]. Cost-cutting exercises, such as the removal of regular maintenance, are often the cause of major faults [7,8]. Conversely, preventative maintenance routinely tests and maintains machinery to detect faults and prolong the health of equipment [5,9]. While this helps to mitigate the negative impact caused by reactive maintenance, this strategy can also be costly in terms of time spent to maintain machinery, unnecessary hardware replacement and factory downtime.

Automating predictive monitoring using IoT and ML is an approach currently being considered to facilitate controlled shutdowns around production as and when they are required. This approach prolongs machine health and reduces the occurrence of unplanned downtime [10]. Combining preventive maintenance with IoT and ML further ensures that machine parts are only replaced when required, which is not always the case in many traditional preventative maintenance strategies [11,12]. The difficulty with ML approaches is the lack of data containing a balanced representation of both normal and abnormal behaviour and the specific classes of fault that exist. More importantly, the performance of many approaches are evaluated under lab conditions.

This paper proposes a state-of-the-art solution to monitor normal rotary machine behaviour obtained from infield motors operating under normal working conditions and highlight the occurrence of anomalous behaviour. Taking into account that anomalous data are a minority, as faults only occur sporadically, the approach taken in this paper focuses on anomaly detection rather than classification to predict when faults might occur. The approach utilises stacked autoencoders (SAEs) to extract features from raw signals found in normal operational behaviour and uses these to identify anomalies (i.e., feature values that deviate from what is considered normal behaviour). The paper primarily focuses on the detection of anomalies in real-time using an edge device, which reduces the overall transmission of data sent for analysis. The edge device detects anomalies and decides what data to transmit for extra analysis or discard. The analysis in this paper is concentrated on electric rotary machines, (Motors, Pumps and Fans), as these have a significant impact on production when a failure occurs [13].

## 2. Related Works

Research are being conducted on the use of machine learning to detect anomalies in the electrical motors across various industries. Machine learning can be used to predict when a motor will fail, allowing for the early detection of breakdown and the preparation of maintenance schedules. Various methods and approaches have been proposed to predict upcoming faults in rotary machines and facilitate predictive maintenance. Manufactures require a system that can distinguish between fault data and known healthy data in real-time [14,15,16]. For example, [17] developed an Automatic Weather Station system (AWS) to collect sensor data and model normal behaviour to reduce downtime in sensor weather stations. The approach model’s behaviour to identify outliers in the sensor data produced in the AWS system using anomaly detection. The abnormal readings were modelled using correlations in data trends and Successive Pairwise REcord Differences (SPREDs). A similar approach was used in [18], whereby sensors are used to monitor devices to measure power consumption, performance and utilisation metrics, to identify abnormalities. Both methods look at current and past trends, which is observed as normal healthy data, and seek to determine a change or anomaly to define an abnormality in current trends.

In the case of predictive maintenance, the data are often asymmetric as discussed in [19]. The main concern highlighted in the paper is the bandwidth needed to continuously transmit real-time sensor signals. To address this, the paper proposes the anomaly detection-based power saving scheme (ADEPOS). Modelling healthy data and detecting changes over time is performed at the edge while only anomalies are returned thus saving bandwidth. Extreme learning machine boundary (ELM-B) is used to detect anomalies and utilises traditional autoencoders (AEs) configured with start-up weights and biases randomly selected from a continuous probability distribution.

Addressing the asymmetric problem and the large amount of data generated by sensors [20] performing all computation on the edge without returning the data. It proposes a fault detection and monitoring system using a long short-term memory (LSTM) to model faults generated by machines in real-time. The approach analyses and detects anomalies at the edge and does not transmit any data back to centralised servers. It is not clear from the paper how the inherent limitation of inferencing complex models on edge devices scales to more complex real-world problems. The paper reports perfect results for Precision and Recall using simulated tests but there is no information on how the model performs when deployed in real-world environments. There is also a fundamental flaw in that the simulated data does not account for the inherent imbalance in machine-generated data under normal working conditions.

AE are more recently being applied to model sensor data. An AE is a feed-forward neural network consisting of an input layer, at least one hidden layer, and finally an output layer. They are typically trained to reconstruct their own inputs by forcing a reduction in dimensionality from the input and therefore have the same number of inputs as outputs. A representation of an AE is displayed in Figure 1, in which the model is used to check sensor data readings and determine an anomaly. Figure 1 displays a representation of an AE with three layers of hidden neurons. An AE with a lesser number of hidden neurons than the input/output layer is forced to learn complex features within the data, as the data are compressed into a lower dimensionality representation of itself, similarly, to learning from principal component analysis (PCA) but from a nonlinear representation. These learnt features are used to reconstruct the input vector and can be used for different purposes, such as dimensionality reduction and image denoising.

SAEs are combinations of multiple pre-trained AEs that are individually trained [21]. By training an initial AE with a single layer, the model takes an initial input, such as a data window, and will reduce or ‘encode’ the input into a much smaller dimensionality, and ‘decode’ this layer to recreate the initial input, as the model’s output. This method allows relationships within the data to be found, by reducing the dimensionality and allowing the model to create its own features prevalent within the data. Subsequent layers can be added to further encode the data and create a more complicated model by creating more condensed features. This allows for the discovery of more relationships within the data and the ability to generalise better.

AEs work well on high-level features but have a limited effect on signals which exhibit complex features. SAEs on the other hand allow more complex features to be extracted. Features present within the data can be found without the need for expert knowledge in signal processing. In [22], SAEs are implemented to determine the health of bearings using the NASA bearing dataset [23]. The results show that the method could achieve 100% detection for bearing faults. This is a promising result as it reduces the need for experts and opens up possibilities for real-time maintenance. Conditioning normality on the data allows the system to detect changes in the signal, which would otherwise be unnoticed when deploying classification.

In paper [24], the author discusses the compute capability of small form factor devices and the benefits of edge computing. The traditional method involves sending all observations to a central server, thereby increasing bandwidth and connectivity requirements. However, the compute capability of these edge devices is ever increasing with annual improvements to the hardware. The author discusses the benefits of deploying an edge-computing model and how edge devices allow analytics to be executed locally and make decisions about observations. This paper emphasises that anomaly detection can reduce the overall bandwidth and connectivity requirements compared to using the traditional data centre-based architecture.

Industry 4.0 and the idea of connecting sensors are a relatively new initiative. Combining this with ML is even newer. The idea of connecting edge-computing devices with data to improve the efficiency of industrial processes is fairly recent; however, edge computing was developed years ago and is now starting to become mainstream. Nonetheless, the background highlights a need for more complex analytics to optimise workflows and maintain the uptime of machinery used in manufacturing processes. This paper builds on the body of work in this area, presents the results of an industrial trial, and assesses the applicability of ML models and their ability to detect anomalies.

The remainder of this paper is organised as follows. The methodology is presented in Section 3, which includes a discussion on data collection, data pre-processing and the configuration of the SAE along with the evaluation metrics used in the experiment to calculate the reconstruction error for anomaly detection. The results obtained from the model are presented in Section 4 and discussed in Section 5, before the paper is concluded and future work is presented in Section 6.

## 3. Materials and Methods

A study was conducted in partnership with Central Group PLC in Liverpool UK. The dataset contains, time-series sensor data, provided by a large steel manufacturer. The data were obtained from a critical machine used in their production line and contain both normal and fault data.

### 3.1. Data Collection and Description

The sensor data were obtained from an electrical rotary machine that adds tension to feed the product through production. The machine is a critical component and therefore has been pre-installed with sensors to determine the speed, current and voltage of the motor in real-time.

The data provided include a large proportion of known healthy data. The manufacturing company also provided fault data, with their engineer reports diagnosing the reason for the failure. The data collected from rotary machines differ depending on the direct application of the equipment. Some rotary machines will have access to a drive, called a drive system, this will allow measurements to be taken such as: speed, frequency, current and voltage. A drive is an electronic device that regulates the electrical energy delivered to a motor. The drive provides the motor with various amounts of electricity and regulates the frequency to control the motors speed.

Measurements taken from the motor include current, status bits, 1st up fault, 2nd 16 faults, armature current reference, actual armature current, speed reference, actual motor speed, field current reference, actual field current and actual armature voltage. Each of the 11 data channels is sampled at 100 Hz and sample readings can be seen in Figure 2.

For the model, a single measure of sensor data was used, actual field current, as this was most consistent across the whole of the normal data and was the single most prominent feature indicating the upcoming fault, as seen in Figure 2. Note that Field Current Ref also shows significant change during the fault indicating that this channel strongly correlated with Active Field Current as shown in Figure 3. As such, we decided to use Actual Field Current only.

Figure 1 shows the relationship between the gathered sensor data. When referring back to Figure 2, the correlation between many sensors is high, close to −1 or +1 showing an almost perfect relationship. Omitting repeated readings is necessary for a stronger more robust model. The repeating readings allow the model to build a more accurate representation of the training data, and therefore have a higher chance of correctly identifying anomalies.

Future data collection will include the edge device collecting sensor data and data from multiple sources. This will be combined with edge modelling. This data will be trained at an offsite PC, using a GPU, by sending the healthy training data. Once the model is embedded, the model will be predictive of abnormalities and will be used to generate a maintenance schedule, by predicting when the motor may need to be inspected.

### 3.2. Data Pre-Processing

The data pre-processing begins by dividing the data into two sections, which includes pre and post-failure. Post-failure data represent the best possible normal running of the motor, as they were subject to a full service and repair. The primary focus is the detection of catastrophic failure. Figure 2 shows two hours of sensor monitoring. The data consist of 1 h 45 min of normal operation (no fault) followed by 15 min of pre-fault data as highlighted in the red box. During the last 15 min, it was clear to see distinct changes in Actual Field Current prior to the actual breakdown which occurred at ~9:20 am. All other signals showing limited variation as such are discarded.

Actual Field Current (AFC) has been identified as a prominent channel of data that can best capture both normal and abnormal operation, for this model. For this study, AFC data from normal operations are extracted for machine learning modelling. This is best displayed by Figure 2, in which the prominent feature throughout is AFC, which is visible by the eye. This is the fault signature for the first anomaly that can be seen to degrade significantly in the graph. To test this theory, each dataset was modelled separately, with AFC detecting the anomalous behaviour significantly earlier than any other sensor channel, and therefore was chosen for this paper.

The time-series signals are segmented into various window sizes and used to model normal behaviour and detect anomalies. The number of hidden node parameters exponentially increases based upon the initial window size, therefore no more than 1000 samples (10-s window) will be used for analysis. In this study manual feature extraction is not implemented. Each of the windows is inputted through an ML model and features are extracted automatically as discussed in the next section. Abnormal behaviour is detected efficiently by a SAE applied using a window model. This was explored to demonstrate the potential of the framework to detect incipient faults in industrial equipment, and provides a foundation on which future work can be built. The framework will include anomaly detected behaviour feeding into fault-detection to advise planned maintenance or repair to avoid potential downtime. Larger window sizes would allow more data to be processed, however the scalability of the model and the model size would suffer as a result. This includes posterior reconstructions, which require considerably more parameters than an anomaly detection model on an edge device model. Therefore, by analysing a window containing a number of seconds of data, the model is able to make accurate predictions with a relatively small number of parameters. Therefore, by applying multiple window levels to the data, we are able to model the anomaly detection and the architecture to begin the classification process.

### 3.3. Machine Learning and Modelling

The framework aims to identify abnormal behaviour in rotary motors using the Actual Field Current signal. The signal is fed into an anomaly detection model and abnormal behaviours are identified by calculating the reconstruction error. Anomaly detection can use the reconstruction capabilities of a pre-trained ML model, to identify when sensor readings become abnormal and therefore could contain valuable information.

The overall process applied in this paper seeks to denoise the sensor data based on known healthy data, and in the process determines anomalies from abnormal readings when new untrained data are tested using an AE. The abnormal readings will give further insights to possibly identify when a sensor reading starts to become abnormal based on known normal data.

The architecture of an AE consists of an encoder and a decoder. The encoder reduces the dimensionality of the data by learning patterns and features of the data. The connections between the input and hidden nodes are the encoding layer of the AE. The decoding layer is the connection between the hidden layer and the output nodes, this segment of the AE uses the features learnt to recreate the input from a lower-dimensional representation. An AE consists of neurons, and these neurons take inputs and apply weight and bias to each input, then aggregated and applied to an activation function, such as a sigmoid function.

Neuron computational formula for updating each weight to approximate the input is defined as:(1)$$h_{W,b}=f\left(\sum_{i=1}^{m}W_{i}x_{i}+b\right)$$
where *W* is weight, *b* is bias, *i* is the number of input neurons, *m* is the max input neuron, *x* is the input vector and *f* is the activation function.

Subsequent neurons learn smaller (dimensional) representations of the inputs by adjusting the weights to reconstruct each input with a closer approximation as these weights are updated. The result allows an N-dimensional dataset to be reconstructed using P neurons as a representation of the dataset (N > P). The AE allows an edge detection model to be inferenced in real-time to determine any abnormal behaviours from the rotary machine.

Backpropagation is used to minimise the overall cost function throughout the network, here this is the overall reconstruction error using mean squared error. This is achieved by optimising the network and calculating the gradient of the cost function with respect to the weights and biases, then adjusting the weights and biases at each layer to improve the overall reconstruction error. Backpropagation trains the weights within the network by optimising each weight based upon the training sample at time *t*, with weight $\omega_{ij}$, and is updated using Equation (2).
(2)$$\Delta \omega_{ij}^{t}=-\epsilon \frac{\partial E}{\partial \omega_{ij}}+\alpha \Delta \omega_{i,j}^{t-1}$$

The network is individually trained layer by layer. This means that backpropagation was not conducted on the final model, as each layer is trained individually. Backpropagating across the whole network would produce a denoising AE (DAE) rather than a SAE, and this is an important distinction to make between the two methods. Figure 4 outlines the method for producing individual models to create a SAE from individual AEs.

Figure 4 displays an example of a SAE model, in which the models are individually trained, and the weights from each model are gathered and collated to create a final model. This architecture is different to the DAE, as this model trains using backpropagation throughout the network, therefore each subsequent layer is trained to minimise the overall error. SAEs use separate models to train a single hidden layer, these hidden layers learn features from the input data to recreate the output layer. The subsequent hidden layer is then used as input to train a further model. This results in the first model learning simple features of the training dataset, the second model learns more complex features based on the initial model′s output from the features. This allows a more complex model to be created, as the model forces the data dimensionality to reduce, in turn determining more complex features in the data.

Figure 5 shows the Framework design to reduce the overall storage and transmission of real-time sensor data to only include data that signify a potential issue with the machine. The figure is comprised of the following:Rotary machines of various kinds with IoT sensors attached delivering sensor data to an IoT device (A).IoT device is embedded with a pre-trained stacked autoencoder with a set of rules to investigate anomalies based on the reconstruction of sensor data (B).Abnormality is detected by the IoT device (C).IoT device transmits a window of T-minutes before the anomaly was detected including up to T-minutes after the final anomaly is detected by the threshold (D).

The “window of T-minutes” before the anomaly is computed, in Figure 5, is a key component to this anomaly detection approach. Once an anomaly is detected, a small window before the anomaly is taken as a prediction, and subsequent anomalies will increase this window of data. The theory is based upon the detection of a sudden change in machine behaviour, this will provide a key insight prior to the anomaly, as well as the ability to discard single-window anomalous behaviour, as potential outliers. The window will ensure that all anomalous behaviour is captured and analysed, to mitigate the likelihood of discarding any key behaviour leading to this change in state from normal to abnormal.

### 3.4. Stacked Autoencoder Configuration

The model is trained using different window size configurations. The first model uses a 500-observation window of sensor data equal to 5 s. In this model, the hidden layer contains 300 neurons, which ensure enough structure is kept from the initial data and sufficient features can be extracted. The output from the hidden layer connects to the output layer which contains 500 neurons. This model is trained to approximate the input on the output layer. Training continues until an optimal reconstruction error is achieved. When training in this model is complete the hidden layer (features) is used as input to a second AE which again is trained to approximate the input on the output layer. In the second model, the hidden layer is reduced to 200 nodes. This configuration continues until the final model contains 70 hidden neurons which after training constitute our final set of features (70 features).

Each individual model is built and combined to form an overall model, using the individual AEs to create an SAE. The SAE has an overall layer structure as shown below.
$$\begin{array}{c} \mathrm{Model}\;1:\;500–300–500\\ \mathrm{Model}\;2:\;300–200–300\\ \mathrm{Model}\;3:\;200–120–200\\ \mathrm{Model}\;4:\;120–70–120 \end{array}$$

The final set of features (70) obtained from the SAE is used to reconstruct the output as shown in model 5.
Model 5: 70–50–500

Using a custom loss function, the model can compare the initial window of data (size 500) with the output layer generated in model 5. The loss function for models 1–4 determine the difference between the model input and output and minimises the loss. This is not suitable for model 5, so the model takes both the initial training data and the subsequent features of the model at 70 layer as shown below:500–300–200–120–70

The SAE model with multiple levels found that the multi-level models provide more accurate results, by modelling only healthy data to learn and to predict the data that will be unhealthy. A SAE model with fewer levels did not reproduce the same results of both modelling normal data effectively, whilst being unable to reproduce unseen abnormal data with the same separation as the larger layered model.

The model evaluates the loss by comparing the model′s input and output based on the reconstruction of the data using the 70 features. This is achieved using a squared distance measure as seen in Equation (3). This equation calculates the vector distances to ensure a close reconstruction has been met. Historical healthy data are used to train the model above to recognise features present during normal operation. Once trained, the model will be unable to recognise unusual data points, as such the reconstruction error will be higher. This is used to detect anomalies in machine operation and is defined as:(3)$$\mathrm{SD}=\mathrm{Mean}({\left(A\;-B\right)}^{2})$$
where A is the initial input windows, and B is the total model output. The squared distance measure in Equation (3) is used to train model 5 and compare the original input using the expanded output generated by model 5. This is a custom loss function added to the model, as it does not take the model′s input features as a measure for the loss function, it uses the initial data window to evaluate the loss function, and the corresponding features of these same windows as the input of the model.

The training of SAEs involves individually training autoencoder models, with an architecture identical to Figure 4 with the A-B-A. To minimize the overall error in each individual model, the output was minimised using the distance measure, mean squared error, which is seen in Equation (5). The output from each model was then combined using a stacked autoencoder. The middle layer of the autoencoder was used to represent the features of the input of the following model. This is both the weights and bias from the encoding layer applied to the input data to generate a feature vector to serve as the input of the following model. This process was repeated for each model, Models 1–4, until the final layer, which uses the features built by the previous layers as input to the final layer to reproduce the input layer. The final layer uses the most condensed form of features to recreate the initial input, as seen in Figure 6, as this reduces the size and complexity of the model as the final layers predict the input as directly from its built-up high-level features.

The model takes a window of data $\;\left(1,A\right)$, for example in Figure 6 the window (A) is set to 500 observations, to evaluate the SD loss measure. Then the model evaluates the features within the first layers of the model, this will encode the data window to a condensed set of features representing the initial input. In Figure 6, the features contain 70 individual data points. These high-level features are then used to train an artificial neural network.

The concept comes from the features generated cannot be evaluated against the required outcome, however, the initial window used to generate the features can be evaluated against the model output. Therefore, we can generate the features for each window, and combine this with the raw data window to evaluate the final model architecture. This is displayed in Figure 6, in which the initial data are analysed using the architecture from the previously trained AE when stacked together. Then the overall feature space is taken, with their pairwise raw input, and trained in Model 5, to determine our final layers in the model, i.e., the reconstruction layers. The overall model is generated by extracting each of the encoding layers for Models 1–4 and stacking them with Model 5 to create the final model. This final model is used as the basis for anomaly detection of the system. Anomalies are identified by the feature vectors of the input and output of the final model. A window is identified as abnormal if the output of the final model is significantly different to the input.

### 3.5. Evaluation Metrics

The models were trained using Tanh for both encoder and decoder. The loss value was computed using mean squared error and the optimiser of the model was Stochastic Gradient Descent (SGD). Gradient Descent (GD) is an iterative process to calculate the optimal value of a function by minimising the cost parameter. SGD uses the same process but randomly selects a small proportion of samples from the training dataset and calculates the gradient. Each parameter is updated from SGD by computing a small number of training samples at each iteration. This allows the model to train using a large dataset which is computationally expensive using GD.

The parameters are updated using the following equation:(4)$$\theta =\theta -\alpha \nabla_{\theta }\;J\left(\theta ;x^{\left(i\right)},y^{\left(i\right)}\right)$$

The SGD formula for updating parameters is based upon a small sample of data taken at each iteration where α is the learning rate of the SGD and *x* is the input vector and *y* is the output vector.

The mean squared error (MSE) loss function calculates the average of squared differences between the given input values, and the predicted output values, and minimises the error across the whole training dataset. This ensures a minimum error across the training data set. MSE is calculated by minimising the following formula shown in Equation (5).
(5)$$\mathrm{MSE}=\frac{1}{n}\sum_{i=1}^{n}{\left(Y_{i}-{\hat{Y}}_{i}\right)}^{2}$$
where the *n* is the total number of samples, $Y_{i}$-input vector, and ${\hat{\text{Y}}}_{\text{i}}$-target output vector.

The reconstruction error is vital in determining unusual activity from the sensor readings. Significant differences will highlight data not contained in the healthy dataset. Therefore, this system will act as a filter to remove any unneeded data, when the system is functioning under normal operation, allowing any data outside of this norm to be identified as an anomaly. These data can then be tested to determine a possible diagnosis, by allowing transmission of data to an outside source, for example an engineer.

The initial parameters for ‘unusual’ readings will be based upon the summary statistics of the healthy dataset. This means determining a percentile region, from which the reconstruction of the healthy dataset contains approximately 99.99% of the healthy data. Therefore, this region will cover almost every healthy data point, except for a handful of windowed sensor readings containing much higher reconstruction error. Further boundaries can be set, using a traffic light system, with green, amber and red showing healthy, unhealthy and critical regions successively. This system is designed to be flexible to accommodate stricter analysis if the targeted rotary machine requires special care. The thresholds can be altered to stricter settings, depending on the needs of the individual, which will allow more data to be abnormal, subsequently transmitting more data for the classification of the fault. The threshold is based upon the sampled training data, after the model is trained, being evaluated by the model for the error value. This threshold is a percentile-based figure of 99.95% for the green threshold, and 99.99% for the most extreme upper bound. This allows flexibility in the model, by reducing the threshold, and allows more data to be deemed abnormal, and conversely increasing it reduces the data deemed anomalous.

Finally, observation made during training and testing found that when sensor data are taken at a resting point, a set of hidden nodes within the model converge to predict this resting period. However, this reduces the overall predictive power of the model for non-resting data. To combat this, no resting points are filtered as the model is unable to predict these data points. This ensures that the model has better convergence during normal running periods.

## 4. Results

The experiments in this section use blocks of data (windows) between 250 and 1000 from the Actual Field Current channel during normal operational conditions. Each model in the SAE is independently trained, based upon the previous model′s condensed hidden layer. Hidden layers are fed as inputs into subsequent models.

Data reconstruction for each window is used to determine thresholding scores. This provides the basis of the anomaly detection, as any unusual patterns in the data should be considered anomalies and require further analysis.

As the healthy data are used to train the model, the reconstruction of this data will be precise. Therefore, using this particular model and reapplying this to the healthy training data allows a threshold to be determined based on the reconstruction error calculated. This will be the basis of the anomaly detection, as the threshold will allow only the most extreme cases.

Each window consists of the same upcoming breakdown, with various window sizes. There are four times as many windows analysed in the 250 data window as the 1000 data window. The main comparison was to determine if any particular model outperforms the others based upon the size of the input data window.

### 4.1. Window–500

Figure 7 shows the reconstruction error when the Actual Field Current is run through the SAE model. Using the reconstruction error, three thresholds are set, these provide a safety measure and ensure anomaly data are detected appropriately.

The thresholds in Figure 7 allow regions to be created, based upon the reconstruction capabilities from the SAE. Having three levels for the threshold provides more flexibility, as a small window of data may show a localised anomaly, compared to a period of windows showing anomalies leading to a failure. The red line in Figure 7 shows the upper limit, which provides a suitable starting point for anomaly detection. The green region also allows for safeguarding, so data on the verge of being an anomaly are also detected.

Figure 8 denotes the reconstruction of unhealthy data, which in our dataset occurred on day 19. This data represent information leading up to the fault. The plot shows a 2 h window before the fault occurred. In this instance, the threshold is reached and maintained above the red line. This occurred 70 min before the critical failure and provides engineers with an opportunity to perform predictive maintenance before catastrophic failure occurs.

A window consistently above the green threshold would be a cause for concern, as the region below contains error for 99.95% of all training data. Therefore, the model identified a sudden change behaviour indicating an anomaly has occurred.

As shown in Figure 8 that the signal between 65 and 190 is missing. This is because the production line has been stopped which is a frequent occurrence. In our approach, the data are bypassed by setting the error to −0.001 which distinguishes the data from normal values. This ensures that a stoppage is not detected as an anomaly.

### 4.2. Window–250

The investigation compares further two window sizes, at 250 samples and 1000 samples. The same structure of the model was used for both the 250 and 1000 sample windows, which means a reduction of ~40% between layers to ensure sufficient information is retained, with the loss function evaluating the final layer.

First, the 250-sample window was used to train the SAE. The SAEs were arranged in the same way as the 500-sample window but with the following configuration:250–150–100–75

The main aim was to compare the known fault to determine whether increasing or decreasing the window size adversely affects the overall model performance, or possibly increases the detection. Figure 9 shows the final reconstruction window based on the performance of the SAE for unseen, abnormal data. The model using a data window of 250 produces a similar result to the model trained with a window of 500. The vast number of observations fall within the green region, and these observations are considered normal. Reconstruction error spikes above the red threshold are prevalent, meaning that the machine shows signs of abnormal behaviour, however, there is a standout difference in performance between the two models. Using a 500-window model has shown that the abnormal data is consistently above the green threshold, and regularly above the red threshold, which is desirable over the 250-window model.

As the reconstruction error in the 250-window model was better for the abnormal data, which is not the aim, the hypothesis is that increasing the model initial window size will also increase its ability to determine abnormalities. This also comes with some limits, as the model should be regularly tested to ensure fast detection of anomalies, by increasing the window size, the model will therefore use more samples to determine a difference, as a result, it will predict less often. This will provide a trade-off between the number of predictions made, and the time taken to determine a fault.

### 4.3. Window–1000

The reconstruction plot for the 1000 window model is shown in Figure 10. The plot shows a significant increase in error received when reconstructing unhealthy data. Compared to the previous window, the model has significantly differentiated between the abnormalities, by using 10 s windows of data within the SAE model.

The results show a significant improvement over previous window sizes. The model is much stricter in determining anomalies when compared to both the 250 and 500 data window models. The thresholds are much smaller, and closer. This means the training data are better represented within this model, as the maximum threshold (red threshold) is far more stringent in this model than in previous experiments.

The thresholding values are collated in Table 1 below for each of the varying window sizes:

The thresholds for the detection of the model are constructed from the 99.95th and the 99.99th percentiles, this ensures the majority of the data are well represented within the model. The model is also much stricter in determining anomalies using the 500 data window model, when compared to both the 250 and 1000 windowed models. The thresholds in this model are much smaller, and closer, due to the model reconstructing the data much better when using the 500 data window model.

### 4.4. Reduced SAE (RSAE) vs. Standard SAE Comparison

To compare our models performance, we rebuilt the SAE using the standard model architecture, and compared the reduced stacked autoencoder (RSAE) with the SAE for their performance. We compared the results of the RSAE model to the standard training method, using the same data set, to see the separation between the models in terms of results. This allowed us to see the potential gains that could be made by training the SAE with our model architecture.

To compare our models performance, we rebuilt the SAE using the standard model architecture initially created for the RSAE. This meant utilizing the same encoding layers used in the RSAE model, from Models 1–4, and their corresponding decoding layers, from 4–1. The output from both models is shown in Figure 11. This graph compares the outputs of each model, in terms of the window error created by estimating the input. The RSAE model creates normal healthy data with a lower error, however the full SAE model distinguishes the abnormalities with a higher error, which appears to be desirable. The problems with the full SAE came from the thresholds created from the percentile, the green (lower) threshold created from the full model was significantly higher than expected, this meant that many of the abnormalities found by the RSAE model were classified as normal when modelled using the full SAE model. Whereas, the RSAE model found right away that abnormalities started to occur.

The comparison of models is conducted upon known abnormal behaviour within the sensor data, once they have been split into windows. This is displayed in Figure 8, Figure 9 and Figure 10 upon the windowed data. The data are known to contain fault signatures, therefore the models should identify these abnormalities as soon as they appear.

Figure 12 displays the full SAE model with its calculated thresholds. This model does seem to separate the normal data from the abnormal data; however, the thresholds set classify many of the known abnormal data within the healthy data.

The two models were then compared based on their detection, using the percentile measure to calculate the upper bounds for an error rate to be considered normal. The confusion matrix is shown below in Figure 13.

In Figure 13, the confusion matrix for our proposed RSAE outperforms the full SAE model in all areas. The model detects the majority of true positives correct, with a true positive rate of 0.875, compared to the full SAE model attaining only 0.2497 true positive rate. Our RSAE model also achieved a sensitivity of 0.9985, which is from the one window classified as true with an actual class of false, whereas the sensitivity of the full SAE model was 1.0, due to the model being more conservative in its positive predictions, classifying less windows overall as true. However, the precision of the RSAE model was 0.92, compared to the full SAE model with an accuracy of 0.5242. A comparison of the false negative rate (FNR) gives the RSAE model a false negative rate of 0.1248, when compared to the FSAE, this has a false negative rate of 0.5653.

## 5. Discussion

This study presented a SAE anomaly detection model to identify abnormal behaviour in rotary machines. Based on the results, the best reconstruction capabilities were from the 1000-window model. The model using 500 samples performed well in determining some possible abnormalities, making use of all three thresholds. This was in contrast to the 1000-window model which is more sensitive to very small anomalies in the Actual Field Current channel. The 1000-window model may be more suitable for motors with small deviations in pattern, but still require 24/7 uptime, as the model takes a brute force approach, in which smaller deviations are always significant.

The 500 window allows the model to determine differences in normal and abnormal data, whilst the severity of the abnormality can also be determined. This will allow the model to transmit small segments of data, that may be abnormal. This provides a safety barrier, that can determine upcoming faults from the different thresholds a model uses. Which model to use is a trade-off and is dependent on the sensitivity required. In cases where there is a need for lower sensitivity levels, the 500 models would suffice, this also translates to a 5 s window of sensor data, making it simple for posterior modelling, i.e., after an anomaly is found.

Figure 7 displays the reconstruction error of training data used to train our model for anomaly detection. We created three thresholds, in which the severity of each observed window could be determined, with any window outside of our most extreme normal data observation requiring further investigation. Having three levels of thresholds allows the severity of the observed windows to be determined. The data below the green threshold will be regarded as normal behaviour, and therefore requires no further action. Data windows falling between green and amber thresholds will be seen as abnormal but not urgent. This category provides engineers with an opportunity for undertaking predictive maintenance. Data failing between amber and red is severely abnormal and requires immediate intervention.

The approach provides an important technique for defining thresholds that are personalised to particular rotary motors. This allows our model to be flexible, as the threshold is currently based on percentiles from our normal data rather than fixed-point settings, these often lead to a high number of false positives. More importantly, these thresholds allow us to detect anomalies seventy minutes before breakdown. This provides critical information particularly in production lines that are fault sensitive and will likely have a significant impact on workflows that contain rotary machines.

The threshold values are very important. They are impacted by the window size, as the larger window size generally has a larger loss value, as the model architecture is much larger to maintain sufficient features. In addition, a larger window will mean more data is included in the posterior analysis, i.e., classification, once an anomaly has been detected. Larger windows will create much larger posterior models, therefore we are careful to keep them small to reduce the time to compute as much as possible.

Within this paper, we chose to use the 500-window model for two main reasons. First, the model has been shown to be effective in anomaly detection whilst containing significantly lower reconstruction errors in the training data. This is evident in the thresholds based on the percentile, the 500-window model is significantly lower than the other two models.

The second was based on the practicality of the model. Using a data window of 500, allows 5 s worth of data to be analysed in the model. Increasing this to a 1000 window would increase the amount of data that could be processed; however, it would also increase the amount of time required to process the data, thus making the potential accuracy of the model suffer. We have chosen to use the model with the smaller window size that still provides the expected results to demonstrate the effectiveness of applying the model to anomaly detection in the context of sensor integrated equipment. This allows a reasonable model size to be chosen, as the number of parameters exponentially grows with larger model inputs and hidden nodes. However, the model outperforms the smaller window model in reconstruction error and therefore was the chosen model within this paper.

For the comparison, we decided to compare the difference between using the full SAE setup, using the entire encoding and decoding layers, as these are the traditional methods for reconstructing inputs using SAE. We found that the RSAE and full SAE have comparable spikes in detection, sharing the same information about the increase in error at the same points. The comparison was to show that both models are able to distinguish between normal and abnormal behaviour; however we believe the RSAE model is superior based upon the much closer representation by the normal data, leaving a much smaller threshold, and therefore is able to decipher abnormalities across all three thresholds, right as the machine state changes.

There is an important distinction to be made between the full SAE and the RSAE, the RSAE model utilised 272 k parameters, whilst the full SAE model used 486 k parameters. The RSAE model found the difference between normal and abnormal data significantly more consistent, as it applies the features and decodes them into the output. This is desirable as computing this model through an edge device requires the model to be as small as possible. Reducing the number of parameters was vital, as this allows the device to compute the model easier.

The confusion matrix also gave an insight in to the performance measures of each model, the RSAE model had great ability to detect abnormalities within the data windows containing abnormal data. Having a sensitivity of 0.9985 and a precision of 0.92 gives excellent scope for filtering unwanted data, when the machine is running with normal parameters. The model also had a true positive rate (TPR) of 0.875 which is a good basis of generating positive data windows holding the key information of a change in circumstances within the machines behaviour. The TPR may be increased by reducing the lower threshold, which will provide more true positives, and possibly more false positives. False positives, in this case, may be more suitable as this model is designed to check for abnormal behaviour. Therefore, it would be suitable to increase the number of false positives provided we identified a higher proportion of true positives in the process. This means no data of value will be lost when filtering data classified as normal. This is a process which we believe will be tailored for each machine based on its designated task. The FNR of the RSAE came out to be 0.1248, based on the confusion matrix. This score does show promising signs for a fast detecting model of abnormalities, as we are able to detect the majority of abnormal behaviour, however this could also be further decreased by reducing the threshold.

This flexibility is vital to the overall framework for future developments, as a static value might be suitable, but having to dynamically change this threshold based on the importance of a machine will allow more stringent classification tests to occur. The advantages of this model are the ability to change the threshold to be more or less strict on the abnormalities found. Having flexibility is vital to both the edge detection, with the target industry motors, and the backend to reduce or increase the number of detected windows for data transmission. The trade-off is between the amount of data sent for analysis, against the cost of a machine breaking down. An ideal scenario would be that all data are analysed; however, filtering using the SAE, with a lower threshold will mean less data are analysed, but the cost of analysis through transmission and storage is less significant.

## 6. Conclusions

This paper proposed the use of a SAE to detect anomalies in signals received from rotary machine motors. The SAE provides an efficient data filter and requires no expert knowledge to determine when faults on rotary motors will likely occur. The SAE model provides a flexible system for automatic threshold identification to identify normal and abnormal behaviour. The results are positive, and the windowing strategy clearly shows ranges of sensitivity levels are possible when evaluating deviations from normal operational behaviour. Once anomalies are detected, the fault signatures can be used to guide engineers to undertake maintenance. However, this does require expertise in signal processing and classification.

This paper found that the 500-window model detects abnormal behaviour efficiently, and that this model can effectively be applied to sensor-integrated equipment. This demonstrates the potential of the framework to detect incipient faults in industrial equipment, and provides a foundation on which future work can be built. The framework will include anomaly detected behaviour feeding into fault-detection and a response mechanism to prevent downtime. We were able to prove that a SAE applied using a window model is able to detect abnormal behaviour in the form of incipient faults within the framework.

The comparison found that we are able to model a traditional SAE model to find abnormalities from a model trained to recreate known normal data; however, the model was outperformed by the one proposed within this paper. We show that the separation of normal and abnormal data is possible, without the need to fully decode the encoded inputs back through the AE architecture. Using the features generated at the centre of the autoencoder allows the model to recreate the output using ~56% of the parameters, as it bypasses the decoding layers by estimating the input directly from the features generated. We believe this is a superior model, using the RSAE model, as it is a much smaller model that reconstructed the normal data appropriately whilst showing that there is a vast difference in behaviour within the machine detected when the machine begins to change in state.

This is the foundation upon which future work can be built. The model will include the detected abnormal behaviour, feeding into the fault-detection system to advise on the potential for downtime. Our proposed method provides a robust system for detection of abnormalities, therefore filtering un-needed data in the process. The application of a threshold is also important, as this allows flexibility within the system, reducing the threshold will increase the number of abnormalities found, allowing for a stricter error-detection system. We believe a 99.99% threshold provided an added level of security to the motor, as a window is unlikely to be found to be abnormal within normal working conditions. Multiple consecutive windows above the same threshold indicate that the motor warrants further investigation. This is the first step in building a system to detect incipient faults in industrial equipment, and provides a platform on which further work can be built. This will provide a platform to allow multiple motors to be analysed at the edge, each with their own individual SAE model for anomaly detection, all of which can be analysed using a single offsite PC deploying classification. By waiting for windows to be found, we believe this is a scalable model analysing only the necessary data points.

In our future work, we will extend our anomaly detection algorithms and add new models trained on fault signatures to automatically classify the different types of faults that occur. The classification model will sit idle until data are provided from the anomaly detector presented in this paper. Future testing will be carried out using a simulation rig, consisting of a motor, gearbox and a pump setup, with access to vibration, temperature and drive sensors. The simulation rig will allow numerous faults to be generated and signatures to be recorded and annotated. The identified future research would include the ability to take the raw signals and apply the model to them, calculate the anomaly and any detected anomaly and transmit the encoded signal. This would further push more computing responsibilities to the edge, as the generated features from the raw signals would be central to the anomaly detection and classification. The edge device would be responsible for the initial processing of the raw signals, followed by any pre-processing that is necessary for the anomaly detection model, and finally the final encoding of the signal for transmission. The edge device could transmit the encoded signal, therefore reducing the amount of data points, and finally the application of classification to the encoded signal.

We also would like to test this model upon further machines, and more sensor readings over time, as an alarm indicator for changes in behaviour that could sit embedded upon an edge device, detecting real time changes. This would allow us to tweak our threshold for each machine, and for each sensor reading.

There is an opportunity to include further sensors within the RSAE and test the potential for transfer learning across different motor types and manufacturers. Having a general-purpose model would be useful when using with a diverse range of motors.

It is also worth noting that the abnormal readings will assist in determining regions of sensor data that could be considered abnormal and therefore will define labels when creating further training data after a breakdown has occurred. The importance of this cannot be understated, as the sensor data are unlabelled, and access to a condition-monitoring engineer with considerable experience in signal processing is not appropriate for every single signal diagnosis.

Overall, the proposed RSAE model delivers an effective edge solution for detecting unusual behaviour, without the need for large data management, transmission and analysis. This model will identify erratic behaviour and indicate this to the operator of the rotary machine, with the opportunity to analyse what this anomaly signifies, and how this may affect production in the short term. This model provides a practical solution to determine localised anomalies on edge devices, whilst allowing information-rich data to be detected and used for further analysis.

## Figures and Tables

**Figure 1 sensors-22-03166-f001:**
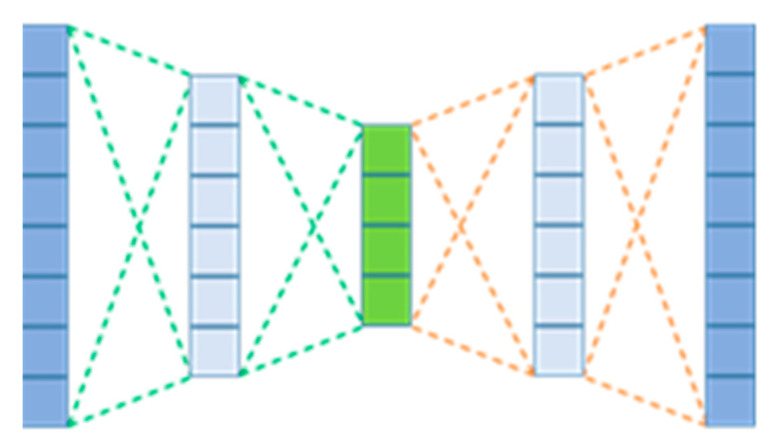
Autoencoder with two encoding layers displayed with green edges.

**Figure 2 sensors-22-03166-f002:**
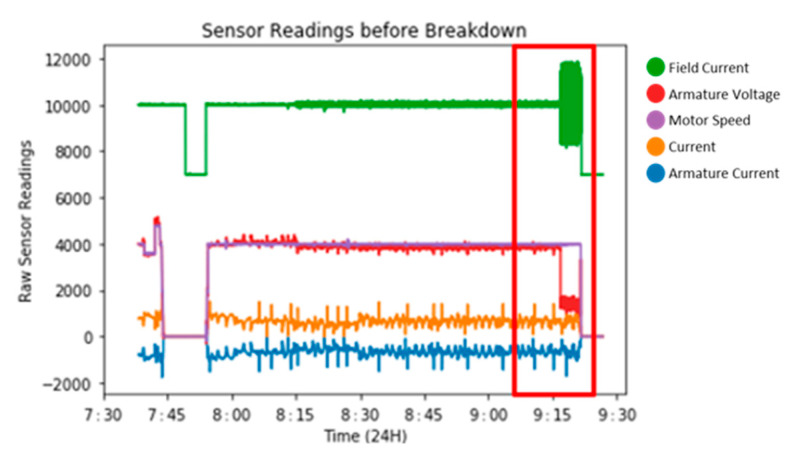
Time series sensor dataset displaying at 1 h 45 min window of data; this is directly before the catastrophic failure highlighted in red.

**Figure 3 sensors-22-03166-f003:**
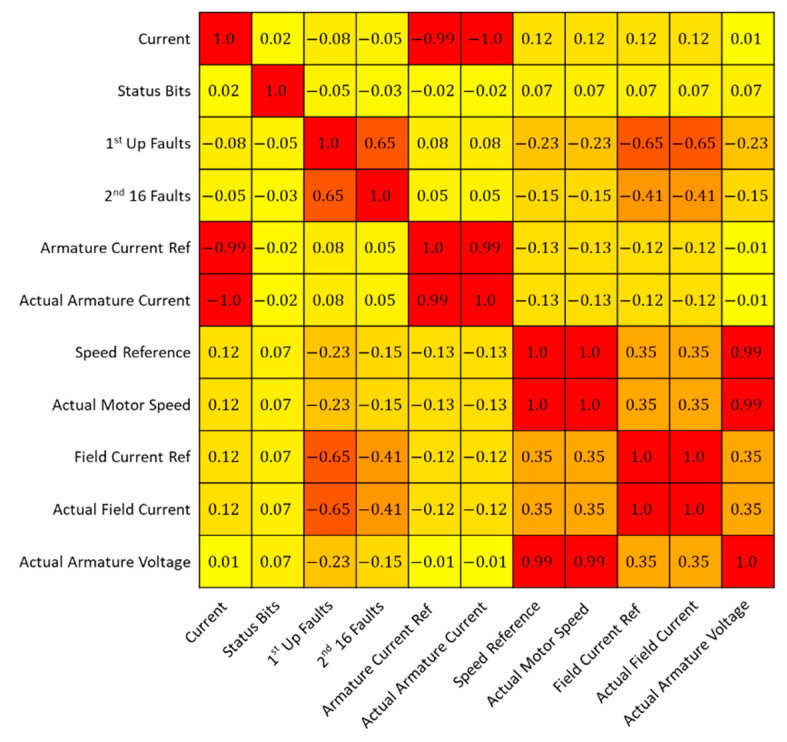
Correlation matrix to test relationships between incoming sensor signals. The colours represent the strength of the correlation, with red representing the strongest relationship regardless of positive or negative, and yellow the weakest relationship.

**Figure 4 sensors-22-03166-f004:**
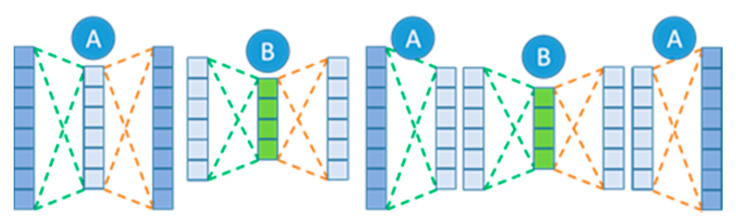
Stacked autoencoder (SAE) architecture, consisting of two single models trained separately, in which the hidden layer of model A being the input layer of model B. To the right, the model is shown in full (A-B-A).

**Figure 5 sensors-22-03166-f005:**
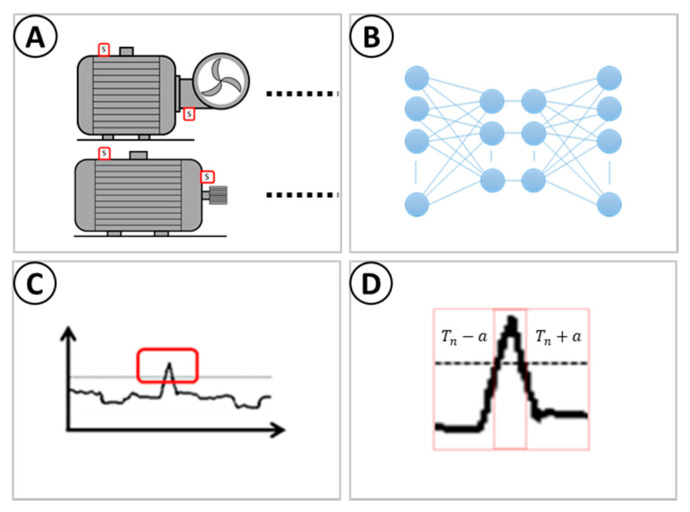
An overall Framework design for anomaly detection on the edge for data transmission. (**A**)—refers to the transmission of sensors to motors, pumps, or fans. (**B**)—refers to the application of a machine learning model to the sensor data, to detect normal running conditions. (**C**)—refers to the detection of a window of data, with an error above normal conditions. (**D**)—refers to the transmission of the window of data, both before and after the abnormality is detected.

**Figure 6 sensors-22-03166-f006:**
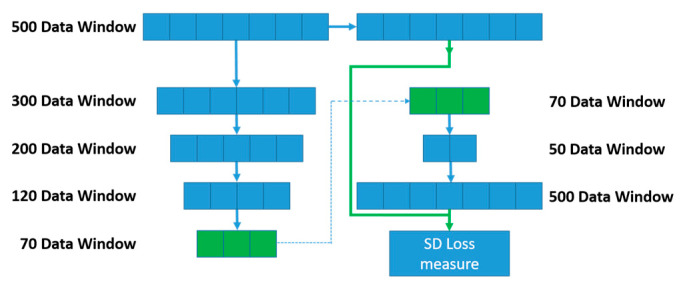
Overview of Model 5, to show the architecture of the model, and how the model evaluates the feature space of 70–50–500, using the initial window of data. Only the model layers 70–50–500 are trained in the final model, this can be described as an artificial neural network.

**Figure 7 sensors-22-03166-f007:**
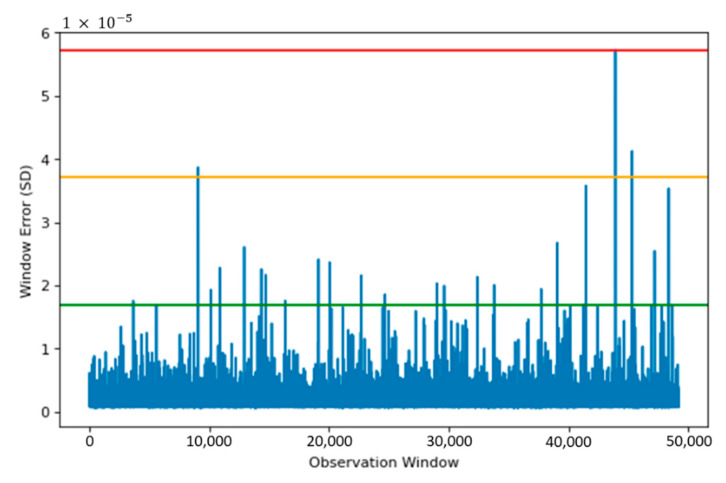
Reconstruction plot using SAE model, on known healthy data, using MSE as the indicator. Red line denotes the upper threshold based upon the 99.99 percentile error from our model. Green threshold is based upon the 99.95 percentile. Orange threshold is the region between the two thresholds.

**Figure 8 sensors-22-03166-f008:**
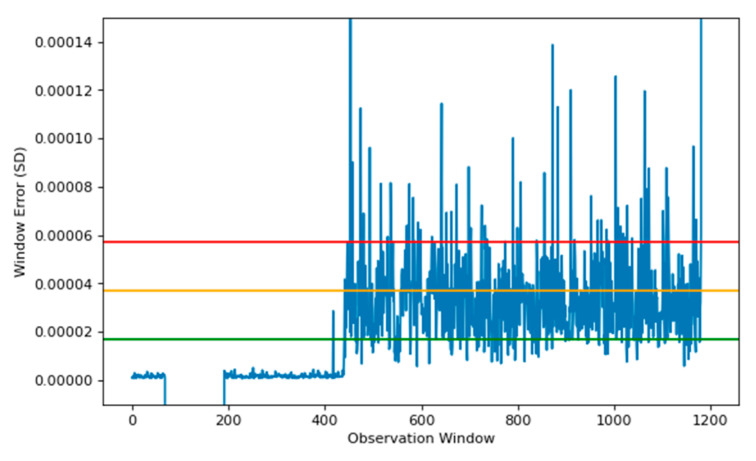
Reconstruction plot using SAE model, using a data window of 500, on known fault data, using MSE as the indicator. The lines denote the previous threshold based upon the healthy data used to train the model. Here the observation window of ~440 indicates a predictive time of 70 min before the initial breakdown occurs.

**Figure 9 sensors-22-03166-f009:**
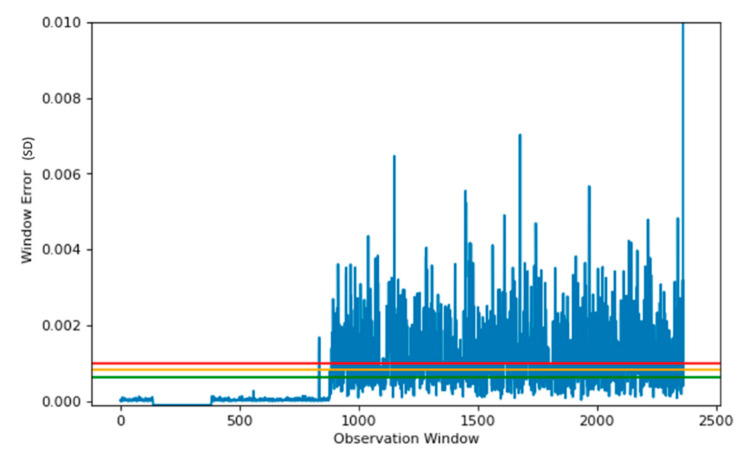
Reconstruction plot using SAE model, using a data window of 250, on known fault data. The lines denote the threshold based upon the healthy data used to train the model, starting at the 99.95% percentile, for the green threshold.

**Figure 10 sensors-22-03166-f010:**
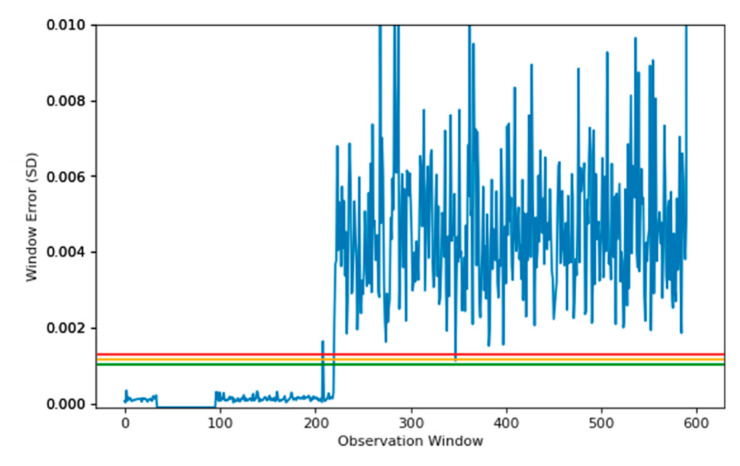
Reconstruction plot using SAE model, using a data window of 1000, on known fault data. The lines denote the threshold based upon the healthy data used to train the model. The same level of thresholds was used in this model.

**Figure 11 sensors-22-03166-f011:**
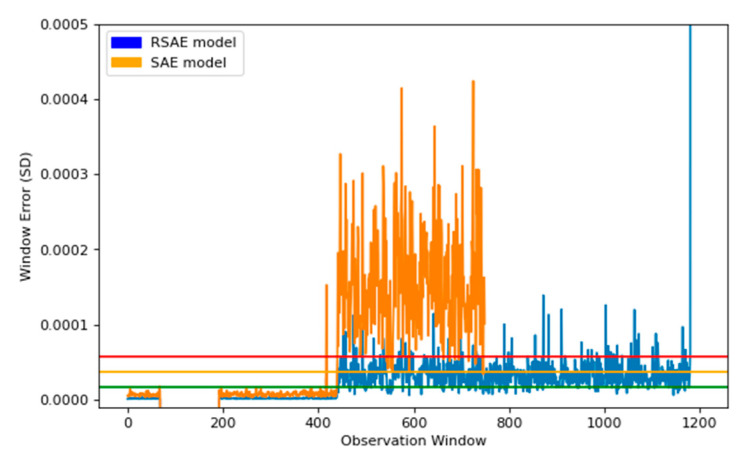
This plot shows the comparison between the RSAE model that we have proposed, and the traditional SAE, shown in orange, using the same encoding layers trained in the RSAE model, and their corresponding decoding layers. The thresholds relate to the RSAE model, shown in blue. The orange SAE model is displayed to 750 windows to illustrate the peaks within both models, but is displayed in full in the next figure.

**Figure 12 sensors-22-03166-f012:**
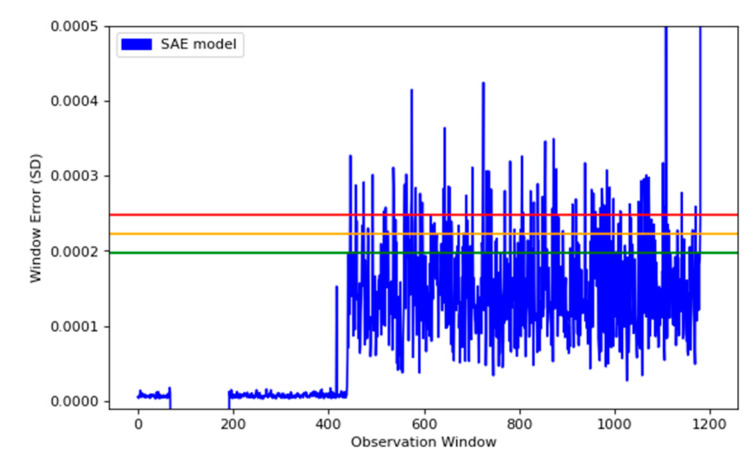
This plot shows the traditional SAE, along with its subsequent thresholds, with green, orange and red being 99.95, ~99.975 and 99.99 percentile error from our model.

**Figure 13 sensors-22-03166-f013:**
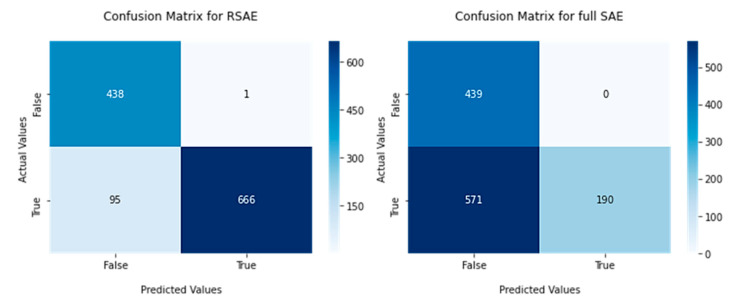
This plot shows the confusion matrix for our proposed RSAE versus the full SAE when applied to data known to be of abnormal behaviour. The RSAE model confusion matrix is displayed on the left, and the full SAE model to the right.

**Table 1 sensors-22-03166-t001:** The thresholds are based on a 99.95% and a 99.99% percentile of healthy training data, for Green and Red. The Orange threshold is the midpoint between the Green and Red thresholds.

Window Size:		250	500	1000
Threshold	Green	$0.652\times 10^{-3}$	$1.686\times 10^{-5}$	$1.04\times 10^{-3}$
	Orange	$0.847\times 10^{-3}$	$3.707\times 10^{-5}$	$1.18\times 10^{-3}$
	Red	$1.022\times 10^{-3}$	$5.728\times 10^{-5}$	$1.32\times 10^{-3}$

## Data Availability

The data presented in this study are available on request from the corresponding author. The data are not publicly available due to privacy and security of data concerned.

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
