# Peer review of "Anomaly Detection Using Autoencoder Reconstruction upon Industrial Motors"

_sensors, 2022, doi:10.3390/s22093166_

Round 1

Reviewer 1 Report

This paper mainly uses the SAE model and AFC signal to automatically detect the abnormal status of the equipment. Except for the method of threshold determination, this paper lacks innovation in theoretical and methodological research.  Moreover, it can be seen from Table 1 that the thresholds obtained are different for different window sizes. In this case, how to choose an appropriate window size to make the determined threshold more match the actual circumstances. This is not clearly stated in the paper. 

In addition, does the method proposed in this paper have advantages compared with other existing methods? 

Author Response

Thank you,

Sean Givnan

Reviewer 2 Report

Dear authors:

This paper applied a Stacked Autoencoder (SAE) model for anomaly detection in signals received from rotary machine motors. Anomaly detection and diagnosis is a very hot topic. The paper is well written.  
The proposed SAE-based anomaly detection approach has been assessed using actual data.  However, several points need more clarification. 

1) This paper is focused on anomaly detection using SAE. Please modify the title to reflect the content of the manuscript. I cannot see 'Real-Time Predictive Maintenance ' in the paper. 

2) The introduction section is long; try to divide it into two parts: 1. Introduction and 2. Related works. 

3) The contribution should be clarified. SAE has already been used in the literature for anomaly detection. 

4) Please try to prevent the idea behind the SAE model clearly with some equations. 

5) The evaluation of the detection performance of the proposed approach must be done in terms of TPR, FAR, accuracy, precision..; otherwise it will be difficult to judge the performance of the proposed model. 

6) Comparison with existing methods is needed to show the benefit of the proposed approach

Author Response

Thank you,

Sean Givnan

Reviewer 3 Report

This paper presents the approach to model normal working operations and detect anomalies in the context of the predictive maintenance.

Comments and suggestions are the following:

  • Currently the theoretical background section does not provide an extensive literature search - lack of current literature in the context of applying different MT techniques to the so-called area: “predictive maintenance”. Could you explain what does mean: “predictive maintenance?” in your approach?
  • Figure 1 – the source is missing.
  • Section 2.2: I am asking for a detailed description of the course of monitored processes and errors. At what time was the research conducted? How was the research system built? Maybe it is worth adding a diagram of the measuring station?
  • Section 2.2 - it should be: “..can be seen in Figure 2” instead of Figure 1.
  • Moreover, a detailed definition of the main concepts of your approach should be added.
  • What accuracy has been obtained using the given ML techniques – could the results be compared?
  • Please define more precisely and detailed the additional contribution of the research to the recent state of the research field.
  • How the findings can be exploited by future similar works?

Author Response

Thank you,

Sean Givnan

Reviewer 4 Report

Comments for the authors:

  • Figure 3 is not clear. What do the authors want to show in figure 3?
  • In section 2.4, why did the author select features number as 70? If some analyses like Normalized Root Mean Square Error (NRMSE) or Akaike Information Criterion (AIC) had been performed to obtain the model order?
  • In section 2.5 the authors have to clarify why they selected 99.95% and 99.99% as thresholds?
  • It should be clarified in the paper why the windows between 250 to 1000 are selected.
  • The quality of the figures should improve.

Author Response

Thank you,

Sean Givnan

Round 2

Reviewer 1 Report

I have no other comments.

Author Response

Thank you for your Review.

Reviewer 2 Report

Dear authors,

The revised version of the manuscript has been improved, but there are still some major points that need to be addressed. 

1) My previous comment has not been addressed appropriately. The 

The evaluation of the detection performance of the proposed approach must be done in terms of  TPR,  FAR,  AUC, accuracy,  precision..;  These metrics are commonly used in this field.  otherwise, it will be difficult to judge the performance of the proposed model.  Also, this will allow comparison with this approach in the future. 

2) In the title, the use of "Real-Time" should be justified in the paper. 

Reviewer 4 Report

My remarks have been addressed properly. I think, it is OK now and can be published.

Author Response

Thank you for your Review.